# Normative Data for Learning and Memory Test (TAMV-I) in Latin American and Spanish Children: An item response theory and linear mixed models approach

Eliana María Fuentes Mendoza[1,2], Laiene Olabarrieta-Landa[1,3], Alberto Rodríguez-Lorenzana[1], Guido Mascialino[4], Esperanza Vergara-Moragues[5], Carlos José de los Reyes-Aragón [6], Natalia Albaladejo-Blázquez[7], Natalia Cadavid-Ruiz [8], María José Irias Escher [9,10], Juan Carlos Arango-Lasprilla[11,12], Erick Orozco-Acosta [13,14]*, Diego Rivera [1,3]*

1 Department of Health Science, Public University of Navarre, Pamplona, Spain, 2 Centro de Experimentación e Investigación Aplicada en Psicometría y Evaluación, CEI-TEST, Universidad Nacional Autónoma de Honduras, Tegucigalpa, Honduras, 3 Instituto de Investigación Sanitaria de Navarra (IdiSNA), Pamplona, Spain, 4 Escuela de Psicología y Educación, Universidad de Las Américas, Quito, Ecuador, 5 Departamento de Psicología, Universidad de Cádiz, Cádiz, Spain, 6 Department of psychology, Universidad del Norte, Barranquilla, Colombia, 7 Department of Health Psychology, University of Alicante, Alicante, Spain, 8 Pontificia Universidad Javeriana, Cali, Colombia, 9 Escuela de Ciencias Psicológicas, Facultad de Ciencias Sociales, Universidad Nacional Autónoma de Honduras, Tegucigalpa, Honduras, 10 Grupo de Investigación en Neurociencias Aplicadas, Universidad Nacional Autónoma de Honduras, Tegucigalpa, Honduras, 11 Department of Cell Biology and Histology, University of the Basque Country, UPV/EHU, Leioa, Spain, 12 IKERBASQUE, Basque Foundation for Science, Bilbao, Spain, 13 Engineering Faculty, Universidad Simón Bolívar, Barranquilla, Colombia, 14 Life Sciences Research Center, Universidad Simón Bolívar, Barranquilla, Colombia

* erick.orozco@unisimon.edu.co (EO-A); diego.rivera@unavarra.es (DR)

## Abstract

Robust normative data for pediatric learning and memory tests in Spanish-speaking populations are scarce, and existing approaches often rely on univariate methods that overlook item-level properties and inter-trial dependencies. The aim was to evaluate the item parameters of the TAMV-I using Item Response Theory (IRT) and to generate covariate-adjusted normative data through Linear Mixed Models (LMM). We hypothesized that the 2-parameter logistic (2PL) model would outperform the Rasch model and that demographic and contextual factors would show significant interactions influencing test performance. The sample consists of 1640 participants from Spain, Honduras, Ecuador, and Colombia. The inclusion criteria were being 6–17 years old, IQ ≥ 80 on TONI-2, and score<19 on the Children's Depression Inventory (CDI). Children with a history of neurological and/or psychiatric disorders were excluded. Item parameters were determined using the 1,2-PL model. LMM were used to evaluate the effect of sociodemographic variables (sex, age, age², mean parent years of education-MPE, country, and interactions). Norms were generated based on participant ability. As a result, the item parameters were calculated and the

**Data availability statement:** All relevant data are available within the paper and its Supporting information files.

**Funding:** This study was funded by the Carolina Foundation for support of the 2024 postdoctoral internship for Carlos José de los Reyes-Aragón. Other authors did not receive any specific funding for this work. The funder had no role in study design, data collection and analysis, decision to publish, or preparation of the manuscript.

**Competing interests:** The authors have declared that no competing interests exist.

LMM showed significant interactions for $MPE * Country$, $Age * Trial$, $Sex * Country$ and $Trial * Country$. By integrating IRT with LMM, this study provides cross-national, covariate-adjusted norms for the TAMV-I, enhancing precision and clinical validity compared to previous approaches.

## Introduction

Children's development relies on several factors, some of them are inherent to the individual, and others, to their environment [1–3]. Children's development is understood as a multidimensional process that includes physical, emotional, social, and cognitive dimensions, allowing the child to adaptively face environmental demands [4,5]. One of these cognitive abilities is learning and memory. Many studies have reported learning and memory difficulties in several neurodevelopmental disorders, such as attention deficit hyperactivity disorder [6,7], specific reading learning disabilities [8], written expression disorder [9], dyscalculia [10], intellectual disability [11,12], and autism spectrum disorders [13,14], among others. These learning and memory difficulties are associated with many functional impairments. Several studies have suggested that children with memory impairments may not only exhibit poor academic performance [15–18], but also social difficulties [19,20], and even some gaps in other cognitive skills, such as language [21].

Given the importance of memory and learning in childhood, many neuropsychological instruments have been developed for children. Some of these instruments were designed for the assessment of an isolated type of memory, such as working memory [22], prospective memory [23], and especially short- and long-term memory [24], in both auditory-verbal [25–27] and visual modalities [28,29]. Other instruments assess memory skills within larger and rigid protocols that evaluate additional cognitive skills [30–32].

Among the most widely used instruments in Latin America for assessing short- and long-term memory are the Rey Auditory Verbal Learning Test, the California Verbal Learning Test, the NEUROPSI, and the Spain-Complutense Verbal Learning Test, while for visual memory, the Rey-Osterrieth Complex Figure Test [24]. However, using these memory assessment instruments in children has some drawbacks. First, most instruments been designed for adult populations from the United States of America and Europe [33]. Although there have been initiatives to develop normative data for other countries [28,34–36], culturally adapted instruments are still limited [33,37]. This inaccuracy in estimating skills could be related, for example, to the type of words included in a word list and their frequency of use at different ages, in various cultures, and even, in different eras [38–40].

Furthermore, it has recently been shown that factors beyond strictly linguistic ones could affect children's cognitive performance, such as parents' educational level [41] and socioeconomic status [32], could affect children's cognitive performance, with the influence of such factors also varying between countries. In fact, according to a study conducted by Arango-Lasprilla et al. [24], which included 808 neuropsychologists

from 17 Latin American countries, more than 60% of the professionals considered the lack of normative data for their country of origin as one of the main problems of assessment instruments. Another disadvantage reported by Arango-Lasprilla et al. [24] study was the high cost of assessment instruments. Approximately 50% of the participants considered the cost of the instruments as a disadvantage of most existing tests. This finding is understandable considering that the average reported income in the study was around USD$1500, and in some cases, a single assessment instrument can cost up to USD$1000.

Currently, one of the main memory assessment instruments for children in Latin America is the Evaluación Neuropsicológica Infantil [Neuropsychological Evaluation for Children] (ENI). This test, developed in Mexico, was designed for Spanish-speaking populations and is, therefore, widely used in Latin America [24]. However, it can be costly for professionals, and its application is time-consuming as it does not only assess memory, but also other cognitive skills and academic abilities [42]. Additionally, the normative data were originally developed for Mexican children, so its use in other countries can lead to biased results. Only one study has developed ENI normative data for the Colombian population [31], but this study was conducted with a sample from a single region of the country. The researchers stratified the sample of 252 children by age, resulting in groups of just over 60 children. Finally, the normative data were generated through multiple analyses of variance (MANOVAs), so the estimates may be less precise than those from more current models [43].

Rivera et al. [44] published a study presenting normative data for the new Test of Verbal Learning and Memory (TAMV-I) for 9 Latin American countries (Chile, Cuba, Ecuador, Guatemala, Honduras, Mexico, Paraguay, Peru, and Puerto Rico) and Spain. This test was developed to assess the learning and memory of Spanish-speaking populations aged from 6 to 17 years, and it has shown good psychometric properties. As an open-license test with normative data for Latin America, it represents a good alternative for clinical practice. However, as with most verbal learning tests, the normative data for total learning scores, delayed recall, and recognition are typically estimated under the assumption that these scores are independent and identically distributed. According to Van der Elst et al. [45], this approach is inadequate when the scores are related, as in the case of the TAMV-I for three reasons. First, the correlated nature of the data is unsuitable for univariate analyses. Second, considering that the TAMV-I yields six scores, statistical models for several of these results need to be tested, potentially increasing type I error, reducing analysis power, and consequently biasing normative data. Finally, calculating six univariate regression models contradicts the principle of parsimony.

From a psychometric perspective, Classical Test Theory (CTT) is widely used, but it assumes constant measurement error across individuals and does not incorporate item-level parameters. Consequently, it provides only limited information about where a test measures most precisely and often relies on separate univariate adjustments for each score, which can inflate error rates and obscure inter-trial dependencies. Moreover, parameter estimates such as difficulty and discrimination in CTT are sample-dependent, which may introduce bias into the results [46]. In contrast, IRT models account for item difficulty and discrimination, allowing measurement precision to vary along the ability continuum. This framework provides richer psychometric information, supports the development of cross-national and covariate-adjusted norms, and facilitates adaptive testing designs [47]. Considering that TAMV-I produces correlated responses across successive trials and delays, combining IRT with linear mixed models (LMM) better captures within-person correlation and between-person covariates than univariate adjustments alone [48]. This methodological approach is not only statistically robust but also clinically meaningful, as it reflects the inter-trial dependencies that clinicians use to interpret performance patterns. In practice, the derived norms reduce the risk of over- or underestimating impairment, thereby improving diagnostic accuracy and guiding more targeted interventions.

To our knowledge, few studies have combined IRT with LMMs to produce cross-national, Spanish-language pediatric norms for list-learning tests. Therefore, this study aims to develop normative data for the TAMV-I test by combining item response theory (IRT) models with mixed-effects models, leveraging the strengths of both approaches to provide robust and precise normative estimates.

## Methods

### Participants

The original sample consisted of 1,748 children and adolescents from Spain ($n=399$), Honduras ($n=288$), Colombia ($n=457$), and Ecuador ($n=604$). Most of the sample were female (52.56%) with an average age of 11.19 (SD = 3.36) and the mean parental education (MPE) was 13.32 years (SD = 3.87). The final sample used for the analyses comprised 1,640 participants with complete data. The sample size for each country was subject to availability at the collaborating institutions rather than predetermined. Nevertheless, local research teams ensured balanced distributions across sex and age groups, and MPE was monitored in each subsample (see Table 1). Following Innocenti et al. [49], and assuming a 95% ($Z_{1-\alpha/2}$) confidence level and with $Z_0 = -0.954$, the expected standard error ($\tau$) was 0.2679 for Spain, 0.3153 for Honduras, 0.2503 for Colombia, 0.2178 for Ecuador, and 0.1280 for the total sample. These values fall within adequate ranges, confirming that the achieved precision is sufficient both for the study aims and for the generation of robust and clinically meaningful normative data. Further details of the sample are available in Table 1.

**Table 1. Sample distribution by country, age, MPE, and sex.**

| Age range (Years) | Age (Years) | | MPE | Sex | |
|---|---|---|---|---|---|
| | n | M (SD) | M (SD) | Girl | Boy |
| | | | | n (%) | n (%) |
| **Colombia** | | | | | |
| 6–8 | 121 | 7.0 (0.8) | 12.3 (3.6) | 63 (52.1%) | 58 (47.9%) |
| 9–11 | 117 | 10.1 (0.8) | 12.3 (3.7) | 63 (53.8%) | 54 (46.2%) |
| 12–14 | 109 | 13.0 (0.8) | 12.3 (3.9) | 59 (54.1%) | 50 (45.9%) |
| 15–17 | 87 | 15.9 (0.8) | 12.4 (4.2) | 50 (57.5%) | 37 (42.5%) |
| **Ecuador** | | | | | |
| 6–8 | 142 | 7.0 (0.8) | 13.5 (3.5) | 70 (49.3%) | 72 (50.7%) |
| 9–11 | 144 | 10.1 (0.8) | 13.7 (4.0) | 76 (52.8%) | 68 (47.2%) |
| 12–14 | 136 | 12.9 (0.8) | 13.5 (3.6) | 69 (50.7%) | 67 (49.3%) |
| 15–17 | 135 | 16.0 (0.9) | 12.8 (3.7) | 69 (51.1%) | 66 (48.9%) |
| **Honduras** | | | | | |
| 6–8 | 68 | 7.0 (0.8) | 13.2 (3.7) | 36 (52.9%) | 32 (47.1%) |
| 9–11 | 92 | 10.0 (0.8) | 12.7 (3.6) | 46 (50.0%) | 46 (50.0%) |
| 12–14 | 68 | 13.1 (0.9) | 12.0 (4.0) | 39 (57.4%) | 29 (42.6%) |
| 15–17 | 56 | 15.9 (0.9) | 13.7 (3.3) | 33 (58.9%) | 23 (41.1%) |
| **Spain** | | | | | |
| 6–8 | 103 | 7.0 (0.8) | 15.0 (3.8) | 53 (51.5%) | 50 (48.5%) |
| 9–11 | 114 | 10.1 (0.8) | 16.3 (3.6) | 56 (49.1%) | 58 (50.9%) |
| 12–14 | 77 | 12.9 (0.8) | 13.7 (4.0) | 39 (50.6%) | 38 (49.4%) |
| 15–17 | 71 | 16.1 (0.8) | 13.5 (3.5) | 41 (57.7%) | 30 (42.3%) |
| **Total** | | | | | |
| 6–8 | 434 | 7.0 (0.8) | 13.5 (3.8) | 222 (51.2%) | 212 (48.8%) |
| 9–11 | 467 | 10.1 (0.8) | 13.8 (4.0) | 241 (51.6%) | 226 (48.4%) |
| 12–14 | 390 | 13.0 (0.8) | 12.9 (3.9) | 206 (52.8%) | 184 (47.2%) |
| 15–17 | 349 | 16.0 (0.8) | 13.0 (3.77) | 193 (55.3%) | 156 (44.7%) |

*Note:* MPE = Mean years Parents Education.

To be included in this study, participants needed to meet the following inclusion criteria: a) be between 6–17 years old, b) be born in any of the four participant countries, c) an IQ ≥ 80 on the Test of Non-verbal Intelligence TONI-2 [50], and d) a score of <19 on the Children´s Depression Inventory [51]. Participants were ineligible if they reported: a) History of central nervous system disorders with neuropsychological impact (e.g., epilepsy, brain injury, multiple sclerosis), b) alcohol abuse or psychotropic substance use, c) uncontrolled systemic diseases causing cognitive issues (e.g., diabetes, hypo-thyroidism), d) psychiatric disorders (e.g., depression, bipolar disorder), e) severe sensory deficits affecting test perfor-mance, f) intellectual disabilities or neurodevelopmental disorders, g) pre-, peri-, or post-natal complications (e.g., hypoxia, seizures), h) having a score of > 5 on the Alcohol Use Disorders Identification Test -AUDIT-C [52] for participants 12 years of age and older, and j) using psychoactive substances such as heroin, barbiturates, amphetamines, methamphetamines, or cocaine in the last 6 months for participants 12 years of age and older.

### Instrument

**Verbal Learning and Memory Test (TAMV-I).** The TAMV-I is a neuropsychological test that evaluates the verbal learning and memory in children, and it consists of three components: free recall, delayed recall, and recognition. Free recall involves four trials where the evaluator reads a list of 12 words (categorized under clothing, furniture, and body parts) after which the examinee is asked to recall as many words as possible. Delayed recall occurs 30 minutes after the fourth trial, where the examinee is prompted to recall all the words s/he can remember from the previous trials. In the Recognition phase, the individual is presented with a list of 48 words, including the original 12 words, along with 12 semantically related words, 12 phonologically related words, and 12 semantically unrelated words. Scoring entails awarding one point for each correctly recalled/recognized word from the original list of 12 words, resulting in a maximum score of 48 for free recall, 12 for delayed recall, and 12 for recognition [53]. In line with the test manual, all administrations were carried out using paper-and-pencil format.

### Procedure

This study is part of a broader research project aimed at generating statistical normative data for various neuropsychological measures across Latin American countries and Spain. Ethical approval was obtained from the following institutions: the Edu-cation Committee at the International University of La Rioja (Spain); the Ethics Committee for Research in the Health Sci-ences Division of the Universidad del Norte (Colombia); the Ethics Committee for Research of the Universidad Pedagógica y Tecnológica de Colombia; the Ethics Committee for Human Research of the Universidad San Francisco de Quito (Ecuador); and the Ethics Committee for Research of the Master's Program in Infectious and Zoonotic Diseases (CEI-MIEZ, Honduras).

Data collection occurred from 03/01/2016–09/06/2017. Local research teams first established agreements with schools and high-schools in each country. Once authorization was obtained from the institutions, the project was presented to students and their families, who were invited to participate on a voluntary basis. Written informed consent was secured from all parents/guardians and participants aged 12 and older, while written assent was obtained from children under 12. The consent process detailed the study's objectives, participant rights, assessment duration and location, and contact information for the local researcher. Parent questionnaires were reviewed before the assessments, which were conducted individually in schools or universities. The neuropsychological battery lasted approximately 120 minutes and was admin-istered in accordance with the guidelines of each test's manual. Participation was voluntary, with no financial incentives offered. Further details are available in Rivera and Arango-Lasprilla [41].

### Statistical analysis

**Item parameters and ability scores.** To determine the item parameters (difficulty and discrimination), IRT was used. Since the data exhibited a dichotomous nature, both the Rasch model and the Two-Parameter Logistic (2PL) model were employed. Subsequently, the likelihood ratio test was used to compare nested models, and the Bayesian Information

Criterion (BIC) to determine the model that demonstrated the optimal fit that solves the problem of overfitting caused by the number of parameters in the model. The Rasch model operates under the assumption that items fluctuate exclusively according to the difficulty parameter [54]. In accordance with Rizopoulos' [54] notation, the mathematical representation for delineating the Rasch model is as follows:

$$log\left(\frac{\pi_i}{1-\pi_i}\right) = \beta_i + \beta_z$$

where $\pi_i$ is the conditional probability of providing a correct response to the $i$th item given $z$, $\beta_i$ represents the parameter denoting the ease of the $i$th item, $\beta$ stands for the discrimination parameter (uniform across all items), and $z$ is the latent ability. The 2PL model estimates both difficulty and discrimination parameters for each individual item.

Once the best-fitting model was selected, item parameters were used to estimate each participant's ability score ($\theta$). This score reflects the underlying performance level by weighting responses according to item difficulty and discrimination, providing a more accurate measure than raw totals. The procedure was applied separately for Trials 1–4, as well as for Delayed Recall and Recognition, yielding comparable and standardized estimates of ability across all test components.

**Demographic effects.** To examine the influence of demographic factors on ability scores ($\theta$), we applied Linear Mixed Models (LMMs), which are well suited for repeated measures data such as the six trials of the TAMV-I (Trials 1–4, Delayed Recall, and Recognition). LMMs allow the inclusion of both fixed effects (trial, age, age$^2$, sex, MPE, country, and their second levels interactions) and random effects to account for within-subject variability. Importantly, in contrast to univariate regression, LMMs model multiple correlated outcomes jointly, meaning that all trial scores are considered within the same framework [55,56]. This can be defined using the following mathematical expression:

$$\theta_{ij} = \beta_0 + b_{0i} + \beta_1 X_{1ij} + \cdots + \beta_k X_{kij} + \epsilon_{ij},$$

where $\theta_{ij}$ represents the ability score for individual $i$ in trial $j$, $\beta_0$ and $\beta_1$ are the fixed effects, $b_0$ denotes the vector of random effects, and $\epsilon_{ij}$ represents the errors for subject $i$.

The Restricted Maximum Likelihood (REML) criterion serves as a measure to assess the fit of the LMM. It operates by considering the likelihood of data transformed into contrasts and offers the advantage of unbiased estimation of $\sigma$ [57]. It has the advantage of estimating covariance parameters while appropriately accounting for the loss of degrees of freedom in estimation [58].

To select the optimal model containing the best predictor variables for ability scores ($\theta$), sequential replacement selection approach was used, which iteratively replaces predictors to improve model fit. This method is computationally efficient and scalable. These models are then compared, and the optimal one is selected according to the BIC [59].

**Normative data procedure.** To generate normative conversions of the ability scores ($\theta_s$) into percentile values adjusted for demographic factors we used predictions from the final LMM. First, the expected ability scores ($\hat{\theta}_i$) were computed using the final linear mixed-effects regression model: $\hat{\theta}_{ij} = \hat{\beta}_0 + b_{0i} + \hat{\beta}_1 X_{1ij} + \cdots + \hat{\beta}_k X_{kij} + \epsilon_{ij}$. Second, the cumulative probability of the observed ability estimates ($\theta_s$) for participant $i$ was obtained from the standard normal cumulative distribution function. Finally, this probability was multiplied by 100 to obtain the corresponding percentile rank ($PR_{ij}$). To facilitate understanding, the Fig 1 provides a schematic diagram of the statistical procedure used in this study.

All analyses were performed using R Project for Statistical Computing for Windows [60] with the *lme4* [61], *lmerTest* [56], and *ltm* packages [54]. The full analysis scripts are available at: https://github.com/diegoriveraps/tamvi-irt-lmm-scripts

## Results

### Item parameters and ability scores

IRT analyses compared Rasch (1PL) and 2PL models for each trial. Likelihood-ratio tests indicated that the 2PL model provided a significantly better fit than the Rasch model ($p < .001$). Consistently, lower BIC values further supported the

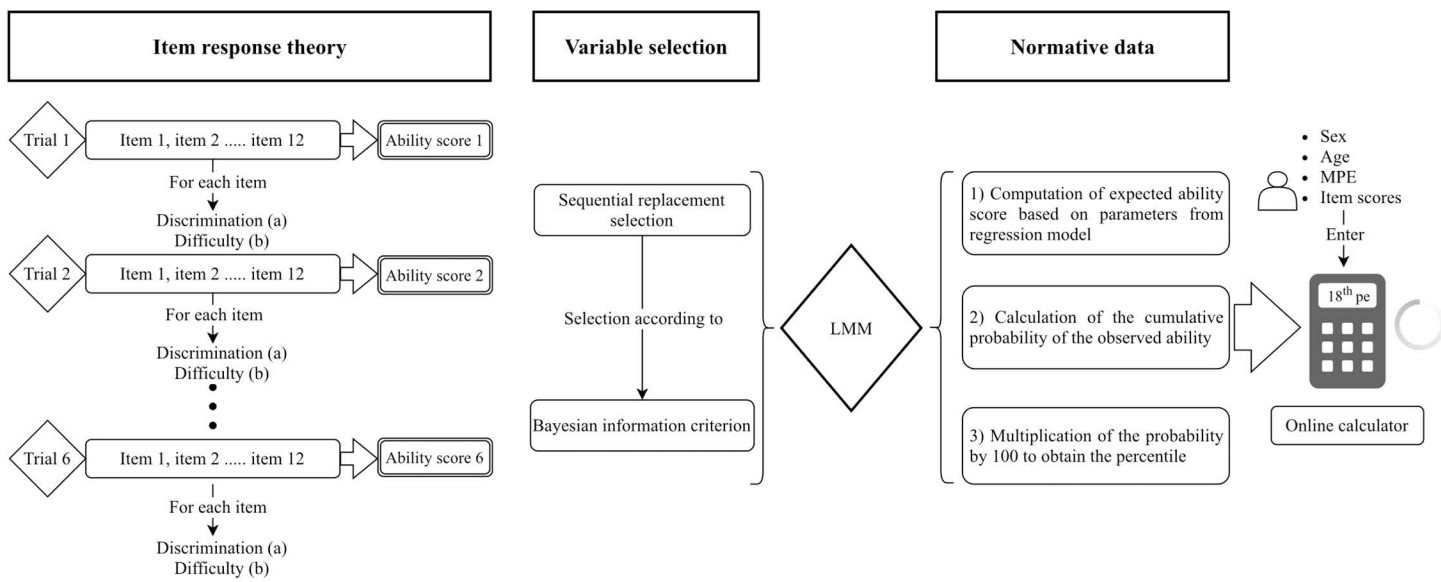

**Fig 1. Workflow of item-level modeling and normative data generation for the TAMV-I.**

selection of the 2PL model, confirming that the additional parameter for item discrimination meaningfully improved model performance. Table 2 summarizes the estimated item parameters (discrimination and difficulty) across the six trials, allowing for the identification of those items that were most effective at differentiating between individuals with varying ability levels, as well as those that were comparatively less informative (for complete results of each trial, see S1 Appendix).

As representative examples, in Trial 4, the easiest item was Nariz [Nose] ($b=-3.17$), whereas Sillón [Armchair] was the most difficult ($b=-0.51$). Discrimination peaked for Zapato [Shoe] ($a=1.09$) and Blusa [Blouse] ($a=1.00$), while Sillón again showed the lowest value ($a=0.30$). In Delayed Recall, Comedor [Dining room] emerged as the easiest ($b=-0.51$)

**Table 2. Most representatively item parameter.**

| Trial | BIC | | Discrimination | | Difficulty | |
|---|---|---|---|---|---|---|
| | **Rasch** | **2PL** | **Upper item** | **Lower item** | **Upper item** | **Lower item** |
| **Free recall-Trial 1** | 25286.57 | 25177.49* | Zapato [Shoe] (a=0.60) | Nariz [Nose] (a=−1.19) | Sillón [Armchair] (b=72.61) | Bufanda [Scarf] (b=−13.68) |
| **Free recall-Trial 2** | 25643.72 | 25560.28* | Armario [Wardrobe] (a=1.75) | Escritorio [Desk] (a<0.01) | Sillón [Armchair] (b=2.20) | Escritorio [Desk] (b=−267.27) |
| **Free recall-Trial 3** | 23521.28 | 23496.06* | Zapato [Shoe] (a=1.39) | Sillón [Armchair] (a=0.33) | Sillón [Armchair] (b=0.35) | Escritorio [Desk] (b=−2.35) |
| **Free recall-Trial 4** | 21005.03 | 21032.14* | Zapato [Shoe] (a=1.09) | Sillón [Armchair] (a=0.29) | Sillón [Armchair] (b=−0.50) | Nariz [Nose] (b=−3.17) |
| **Delayed recall-Trial 5** | 22737.40 | 22761.46* | Blusa [Blouse] (a=1.24) | Nariz [Nose] (a=0.57) | Sillón [Armchair] (b=0.60) | Armario [Wardrobe] (b=−2.04) |
| **Recognition-Trial 6** | 7782.25 | 7824.35* | Ojo [Eye] (a=3.20) | Armario [Wardrobe] (a=1.63) | Sillón [Armchair] (b=−1.60) | Nariz [Nose] (b=−2.45) |

*Note:* BIC = Bayesian Information Criterion; * $p$-value < 0.001 for Likelihood Ratio Test. This table shows which items in each trial were the easiest, the hardest, and the most effective at distinguishing between children based on their abilities.

and Sillón as the hardest ($b = 0.60$), with discrimination highest for Blusa ($a = 1.24$) and lowest for Nariz [Nose] ($a = 0.57$). Finally, in the Recognition trial, Nariz ($b = -2.46$) and Comedor ($b = -2.21$) were among the easiest, whereas Sillón remained one of the hardest ($b = -1.60$). Discrimination values reached their maximum in this condition, with Ojo [Eye] ($a = 3.20$) showing the strongest slope, closely followed by Oreja [Ear], Boca [Mouth], and Bufanda [Scarf] ($a \approx 2.6–2.7$), while the lowest value corresponded to Comedor ($a = 1.97$).

These findings illustrate how certain items are particularly sensitive to differences in ability, while others are more easily accessed regardless of ability level. Fig 2 (left panel: Trial 4; right panel: Recognition) displays the corresponding Item Characteristic Curves (ICCs), highlighting the sharper slopes in Recognition that reflect stronger discrimination. The complete set of parameter estimates for all trials is provided in S2 Appendix. Based on these parameter estimates, ability scores (θ) were calculated for each participant in each of the six trials, which served as the outcome variables in the subsequent analyses examining demographic effects.

## Demographic variables effect

The influence of demographic covariates on children's performance across trials was examined using a multivariate framework. The initial specification of the linear mixed-effects regression model included age, age², MPE, sex, country, and all their second-order interactions. After variable selection process, the final linear mixed-effects regression revealed several significant interactions influencing ability scores (see Table 3). A robust effect was observed for the interaction between *ln*(MPE) and country (Fig 3A): although all countries showed increasing ability scores with higher MPE, participants from Ecuador consistently achieved higher performance than those from the other countries. A second important effect was the age × trial interaction (Fig 3B): in Free Recall–Trial 1, ability scores showed minimal improvement with age, suggesting that

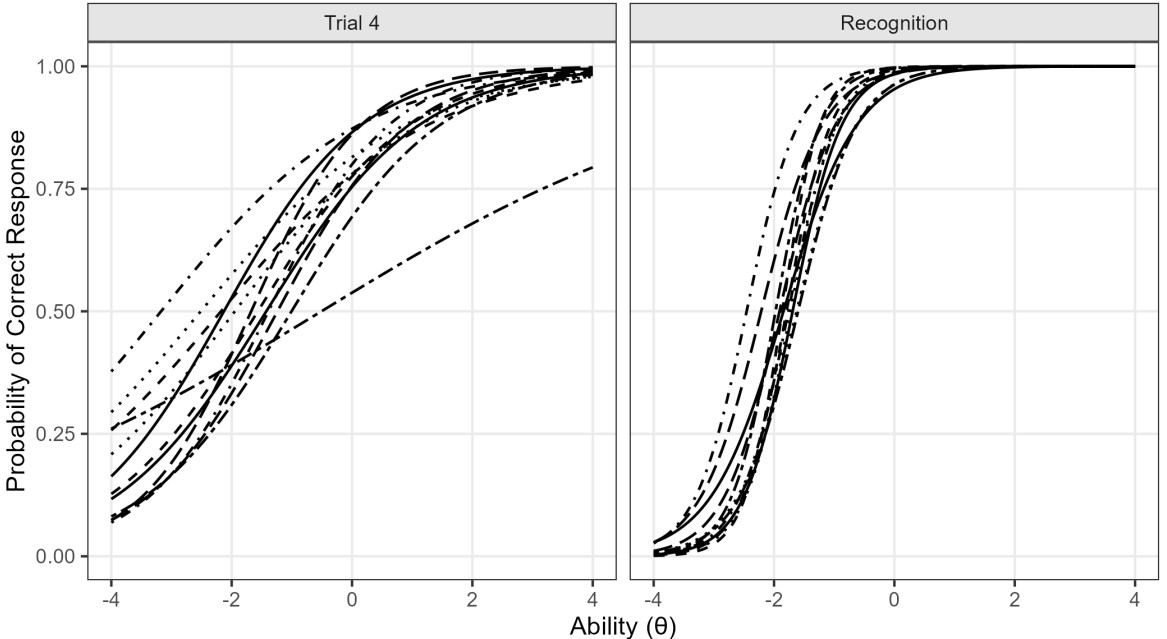

**Fig 2. Item characteristic curves for Trial 4 and Recognition.** Item characteristic curves (ICCs) from the two–parameter logistic (2PL) model for Trial 4 (left panel) and Recognition (right panel). The x-axis represents the latent ability level (θ), with higher values indicating better performance, whereas the y-axis indicates the probability of correctly responding to a given item. Each curve corresponds to one of the 12 items, and its horizontal position reflects item difficulty (b), while the steepness of the slope reflects item discrimination (a). Compared with Trial 4, the ICCs in the Recognition condition are notably steeper, indicating higher discrimination values and thus greater sensitivity in differentiating among individuals across ability levels.

**Table 3. Final linear mixed model.**

| Predictor | Beta | Standard Error | t | p-value |
|---|---|---|---|---|
| Intercept | −0.302 | 0.142 | −2.123 | 0.034 |
| Age | 6.515 | 1.660 | 3.818 | <0.001 |
| Age$^2$ | 0.923 | 1.657 | 0.557 | 0.573 |
| $In$(MPE + 1) | 0.118 | 0.054 | 2.189 | 0.029 |
| Trial 2 | −0.154 | 0.039 | −3.977 | <0.001 |
| Trial 3 | −0.142 | 0.039 | −3.654 | <0.001 |
| Trial 4 | −0.117 | 0.039 | −3.007 | 0.003 |
| Trial 5 | −0.122 | 0.039 | −3.142 | 0.002 |
| Trial 6 | −0.095 | 0.039 | −2.452 | 0.014 |
| Sex (Girl) | 0.023 | 0.041 | 0.567 | 0.573 |
| Country (Ecuador) | −0.020 | 0.199 | −0.098 | 0.919 |
| Country (Honduras) | −0.086 | 0.251 | −0.344 | 0.733 |
| Country (Spain) | −0.265 | 0.279 | −0.674 | 0.342 |
| Age*Trial 2 | 13.909 | 1.982 | 7.017 | <0.001 |
| Age$^2$*Trial 2 | −7.943 | 1.980 | −4.011 | <0.001 |
| Age*Trial 3 | 18.500 | 1.982 | 9.333 | <0.001 |
| Age$^2$*Trial 3 | −9.077 | 1.980 | −4.583 | <0.001 |
| Age*Trial 4 | 15.631 | 1.982 | 7.886 | <0.001 |
| Age$^2$*Trial 4 | −11.924 | 1.980 | −6.021 | <0.001 |
| Age*Trial 5 | 17.740 | 1.982 | 8.950 | <0.001 |
| Age$^2$*Trial 5 | −13.021 | 1.980 | −6.575 | <0.001 |
| Age*Trial 6 | 14.274 | 1.982 | 7.201 | <0.001 |
| Age$^2$*Trial 6 | −15.345 | 1.980 | −7.748 | <0.001 |
| $In$(MPE)*Country (Ecuador) | −0.016 | 0.074 | −0.212 | 0.834 |
| $In$(MPE)*Country (Honduras) | 0.033 | 0.094 | 0.354 | 0.725 |
| $In$(MPE)*Country (Spain) | 0.045 | 0.101 | 0.440 | 0.474 |
| Sex (Girl)*Country (Ecuador) | 0.100 | 0.055 | 1.820 | 0.069 |
| Sex (Girl)*Country (Honduras) | −0.039 | 0.066 | −0.596 | 0.551 |
| Sex (Girl)*Country (Spain) | 0.027 | 0.061 | 0.375 | 0.655 |
| Trial 2*Country (Ecuador) | 0.156 | 0.052 | 3.010 | 0.003 |
| Trial 3*Country (Ecuador) | 0.089 | 0.052 | 1.725 | 0.084 |
| Trial 4*Country (Ecuador) | 0.034 | 0.052 | 0.658 | 0.510 |
| Trial 5*Country (Ecuador) | 0.072 | 0.052 | 1.382 | 0.166 |
| Trial 6*Country (Ecuador) | 0.040 | 0.052 | 0.781 | 0.434 |
| Trial 2*Country (Honduras) | 0.046 | 0.062 | 0.749 | 0.454 |
| Trial 3*Country (Honduras) | 0.132 | 0.062 | 2.137 | 0.032 |
| Trial 4*Country (Honduras) | 0.117 | 0.062 | 1.897 | 0.058 |
| Trial 5*Country Honduras) | 0.119 | 0.062 | 1.925 | 0.054 |
| Trial 6*Country (Honduras) | 0.104 | 0.062 | 1.689 | 0.091 |
| Trial 2*Country (Spain) | 0.420 | 0.057 | 7.309 | <0.001 |
| Trial 3*Country (Spain) | 0.398 | 0.057 | 6.933 | <0.001 |
| Trial 4*Country (Spain) | 0.381 | 0.057 | 6.640 | <0.001 |
| Trial 5*Country (Spain) | 0.346 | 0.057 | 6.027 | <0.001 |
| Trial 6*Country (Spain) | 0.276 | 0.057 | 4.808 | <0.001 |

*Note*: MPE = Mean Parent years of Education; $\sigma_\varepsilon$ = 0.571. This table shows the variables that influence children's performance on TAMV-I.

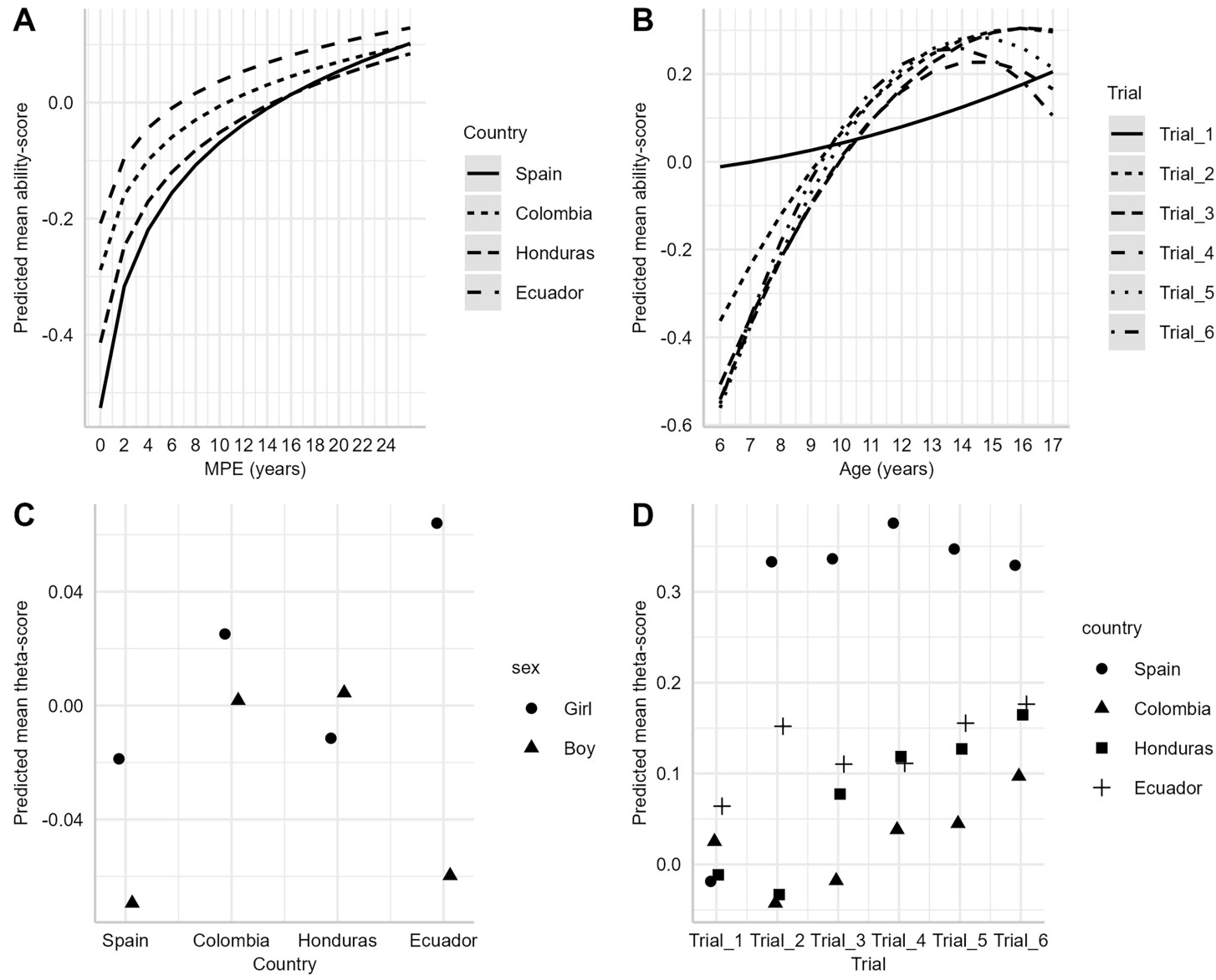

**Fig 3. Final Multiple Linear Regression Model.** Mixed model predictions for theta scores according to (a) years of parental education (MPE), (b) student age, (c) country and sex, and (d) trial and country. The lines represent the mean predicted ability score (theta); a consistent main effect of country is observed (Spain and Colombia outperforming Honduras and Ecuador), while age and trial show smaller differences, and for sex by country where ecuatorian girl has better performance than boys.

age exerted little influence in the initial trial. In contrast, in subsequent trials performance increased until around 13 years of age, after which it declined. A third relevant effect was the sex × country interaction (Fig 3C), which indicated that girls consistently outperformed boys across countries, with the exception of Honduras, and with more pronounced differences observed in Spain and Ecuador. Finally, the trial × country interaction (Fig 3D) revealed that children from Spain generally performed better in all trials, except in Free Recall–Trial 1 where children from Ecuador obtained the highest ability scores.

These results provided the foundation for developing normative data adjusted for each child's demographic background. In practical terms, this means that the ability score obtained by a participant can be directly compared with that

of peers of the same age, sex, country, and MPE. Such adjustments are essential to avoid misleading conclusions, for example, attributing a low score to cognitive difficulties when it may instead reflect differences in age or educational environment. The final normative data derived from these models therefore allow clinicians to interpret an individual child's performance more accurately and fairly within the appropriate reference group.

## Normative data application

As an illustrative case, we considered a 17-year-old girl from Spain whose parents had an average of 18 years of education. Her performance was examined in Trial 6 (Recognition), where she scored 1 (correct) on all recognition items except for Item 1, where she scored 0. Based on this response pattern, and using the 2PL model (including item difficulty and discrimination parameters), the individual ability estimate was $\theta_i = -0.274$. To estimate the normative data for this participant, the procedure described in the Methods section was followed. First, the expected ability score for a participant with the same demographic profile was obtained from the final linear mixed-effects regression model (Table 3), resulting in $\hat{\theta}_i$ = 0.311. Second the observed ability derived from the 2PL model ($\theta_i = -0.274$) was compared against the expected score predicted by the final linear mixed-effects model ($\hat{\theta}_i$ = 0.311). Using the model's residual standard deviation ($\sigma_\varepsilon$ = 0.571), the cumulative probability of obtaining an ability score less than or equal to the observed value was calculated as 0.153. Finally, this probability was multiplied by 100 to provide the normative percentile, corresponding to the 15.3th percentile. In other words, the participant's performance was higher than approximately 15% of peers matched on age, sex, country, and parental education.

Given its complexity, an online calculator has been developed to facilitate clinical practice which is based on the platform https://www.rstudio.com/products/shiny/. It allows for the computation of ability ($\theta$) scores, as well as demographically adjusted z-scores and percentiles. Clinical psychologists only need to input specific patient information requested by the calculator, including item-by-item test responses (1 = correct; 0 = incorrect), age, MPE, country, and sex. This tool is accessible free of charge to all users at https://diegorivera.shinyapps.io/calculator_tamvi_tri/

## Discussion

The objectives of this study were threefold: (1) To evaluate the discriminative ability and difficulty of TAMV-I items across different trials, analyzing the informative contribution of items on ability levels as a function of parameters obtained from the 2PL model, (2) To develop normative data for the TAMV-I using IRT models, and LMM, and (3) To provide a practical and accessible tool for professionals to calculate ability scores, zeta scores and adjusted percentiles for the TAMV-I to facilitate their clinical practice.

Our results confirm that the 2PL model offered a superior fit compared to the Rasch model, supporting the use of IRT as the methodological foundation for TAMV-I normative data. This empirical evidence is consistent with the broader limitations of Classical Test Theory (CTT), which assumes constant measurement error across individuals, relies on univariate adjustments for each score, and does not incorporate item-level parameters. Such limitations can inflate error rates, obscure inter-trial dependencies, and yield parameter estimates that are sample-dependent and potentially biased [46]. In contrast, IRT models, including Rasch and 2PL, explicitly account for item characteristics, but the 2PL model provides greater flexibility by estimating both item difficulty and discrimination, which resulted in a better fit for our data [47]. This framework provides richer psychometric information, supports the development of cross-national and covariate-adjusted norms, and facilitates adaptive testing designs. The analysis of item discrimination and difficulty parameters further confirmed this point, revealing substantial variability among items and underscoring the importance of an item-level approach. For instance, in Free recall – Trial 1, the item Zapato [Shoe] was the best item for discrimination, suggesting its capacity to differentiate between individuals with varying levels of cognitive ability. Conversely, Nariz [Nose] exhibited a negative discrimination value, indicating a reduced efficiency in distinguishing between different ability levels. The difficulty parameters also showed a similar pattern. Sillón [Armchair] presented an exceptionally high difficulty value in Free recall – Trial

1, becoming a highly challenging item. In contrast, Bufanda [Scarf] showed a very low difficulty value, and, therefore, extremely easy item.

Some items showed notable variations in their discrimination and difficulty parameters. Zapato [Shoe] exhibited consistent moderate to high discrimination values, indicating its reliability in distinguishing between different levels of cognitive ability across trials. Sillón [Armchair] showed significant fluctuations in difficulty, ranging from highly difficult in Free recall – Trial 1 to progressively easier in subsequent trials and Delayed Recall. Nevertheless, it remained the most difficult item in each trial. Nariz [Nose] consistently had negative or low discrimination values, as well as low difficulties values, suggesting its poor capacity to differentiate between individuals because it is an easy item.

Such disparities were consistently evident across subsequent trials, with items like Armario [Wardrobe] in Free recall – Trial 2 demonstrating high discrimination and items like Escritorio [Desk] in the same trial exhibiting extreme ease. Although the words for the TAMV-I were selected based on word frequency in Spanish, calculated from both Spanish and Cuban samples [53], the observed variability in item discrimination and difficulty could be attributed to differences in word frequency and familiarity for the children participating in this study.

Differences were also observed between Free recall trials, Delayed recall, and Recognition tasks, and interestingly, as trials progress, the scores showed better parameters. This may reflect the learning process of the person. This finding aligns with the results reported by [62] in their research on the California Verbal Learning Test – Second Edition (CVLT-II). Also, Free recall trials, particularly the earlier ones, might be influenced by learning effects and potential fatigue, as evidenced by the changing discrimination and difficulty values of items like Zapato [Shoe] and Sillón [Armchair]. The increase in discrimination values for Zapato [Shoe] from Free recall – Trial 1 ($a = 0.60$) to Trial 3 ($a = 1.39$) suggests that participants improved their ability to recall this item with repeated exposure. In delayed recall, on the other hand, items such as Blusa [Blouse] maintained high discrimination ($a = 1.24$), indicating effective differentiation even after a delay, while Nariz [Nose] showed low discrimination ($a = 0.57$), highlighting its reduced efficiency in delayed contexts. This balanced mix of item difficulties and discrimination ability is essential to effectively span the spectrum of cognitive abilities being measured.

Analysis of the effect of demographic variables using LMM revealed significant interactions showing the effect of factors such as age, MPE, sex, and country of origin on the learning and memory skills assessed. As observed in Van der Elst's [45], analysis of the Rey Auditory Verbal Learning Test (RAVLT), the age of participants influenced performance across all trials. This interaction showed expected patterns associated with maturation processes of the brain [63]. Interestingly, for Free recall – Trial 1, the required skill level was high and performance increased slowly with age. This could be because participants faced first trial without the familiarity or practice acquired in later trials, suggesting that this trial may be novel, complex, and measuring a different cognitive domain than the other trials, primarily attentional ability. In the later trials, while this trend in performance was also observed, at 13 years there is a plateauing and subsequent decline, suggesting a typical cognitive developmental curve. These findings are in line with previous literature that identifies the peak of cognitive development in early adolescence followed by a stabilization or mild decline [64,65].

The interaction between MPE and country showed that, in general, as expected, higher parental years of education is associated with better performance on the TAMV-I. Higher MPE is usually associated with a more cognitively stimulating family environment, which could facilitate the development of learning and memory skills as suggested by multiple studies [66–68]. The outstanding improvement in children from Ecuador with high levels of MPE (see Fig 2), suggest that, in this country, the benefits of high parental education may be more pronounced than in the rest. A possible explanation for these differences could be that parental stimulation at home, informed by parental education, is more impactful in Ecuador due to a less effective educational system influenced by various socioeconomic and cultural factors within the country [69].

The sex by country interaction reflected better performance of girls compared to boys in almost all countries except Honduras. This finding is consistent with previous research indicating that girls tend to score higher on tests that assess verbal and memory skills than boys [70,71]. The most pronounced differences, observed in Spain and Ecuador, could be influenced by cultural factors, such as gender expectations and/or parenting styles that favor the development of verbal

skills in girls [72]. At the other hand in the case of Honduras, women face several sociocultural disadvantages that impact their performance, these challenges include a higher rate of illiteracy and the traditional expectation that their duties are primarily focused on domestic tasks [73].

Regarding the trial by country interaction, Spanish children performed better, with the exception of the first trial of the test, where children from Ecuador outperformed children from other countries. This result suggests that there may be differences in task preparation and familiarity between countries. The higher scores of Spanish children on trials 2–6 could be due to greater exposure to educational practices that emphasize learning and memory skills from an early age. On the other hand, the higher performance of Ecuadorian children in the first trial could indicate differences in motivation or initial approach to tasks.

This paper presents with several strengths. Firstly, the analyses were performed on a large sample from various countries in Latin America and Spain, enhancing both representativeness and generalizability. Secondly, the study employed a hybrid approach combining IRT and continuous-norming techniques, which allowed for greater precision by leveraging IRT-derived ability scores and adjusting for demographic factors, including country of origin, via regression analysis. Thirdly, the use of regression-based norms allowed us to control for demographic variables related to cognitive performance, and therefore the normative data produced is applicable to populations with demographic differences captured in the regression equation. To the authors' knowledge, the methods applied in this study have been rarely used despite its benefits. Additionally, an accessible online calculator is provided at https://diegorivera.shinyapps.io/calculator_tamvi_tri/.

However, the study also has its limitations. While linear and quadratic models were tested, other polynomial models (such as cubic or logarithmic functions) were not explored, which might have improved model fit, but deviated from the principle of parsimony. Moreover, the study could have included additional variables, like socioeconomic status, quality of education. These aspects could be considered in future research, although most normative data studies use sex, age, and education because these variables are more easily standardized across different test administrations and populations. In addition, while the normative data provided here are robust for the populations sampled in each country, they should not be assumed to capture the full variability of all subgroups. For instance, ethnic and racial minorities may present distinct cultural or educational experiences that can significantly impact their performance on certain assessments. Therefore, clinicians are advised to apply the norms with caution in such subpopulations, as relying on generalized data may reduce diagnostic accuracy for these specific groups [74].

The study's findings have significant clinical implications. Normative data in Latin America are scarce compared to the extensive data available in the United State of America and European countries. Enhancing normative tools for under-represented populations is likely to advance neuropsychological practice by providing more appropriate reference populations. Additionally, using a distribution of theta scores instead of true scores can increase precision, thereby improving diagnosis and treatment. Considering and controlling for the impact of demographic variables when deriving scores will also enhance precision, especially in Latin American countries with significant demographic disparities.

## Conclusion

The present study provides robust normative data for the TAMV-I using IRT and LMM models. The choice of the 2PL model, based on the BIC fit indices, over the Rasch model, demonstrates that this model allows a superior fit for the test data. In addition, the findings obtained reflect the importance of considering demographic and contextual variables in the interpretation of the results. Parental education, age, sex and country of origin were shown to have a significant influence on the scores obtained by the participants. Finally, the online tool and normative data developed in this study represent a valuable contribution to clinical practice, facilitating accurate and accessible score calculations for psychologists.

## Supporting information

**S1 File. Anonymized dataset.**
(CSV)

## Author contributions

**Conceptualization:** Laiene Olabarrieta-Landa, Erick Orozco-Acosta, Diego Rivera.

**Data curation:** Eliana María Fuentes Mendoza, Esperanza Vergara-Moragues, Natalia Albaladejo-Blázquez, Natalia Cadavid-Ruiz, María José Irias Escher.

**Formal analysis:** Eliana María Fuentes Mendoza, Erick Orozco-Acosta, Diego Rivera.

**Investigation:** Alberto Rodríguez-Lorenzana, Guido Mascialino.

**Methodology:** Diego Rivera.

**Project administration:** Juan Carlos Arango-Lasprilla.

**Supervision:** Laiene Olabarrieta-Landa, Juan Carlos Arango-Lasprilla, Diego Rivera.

**Visualization:** Erick Orozco-Acosta.

**Writing – original draft:** Eliana María Fuentes Mendoza, Laiene Olabarrieta-Landa, Alberto Rodríguez-Lorenzana, Guido Mascialino, Carlos José de los Reyes-Aragón, Diego Rivera.

**Writing – review & editing:** Laiene Olabarrieta-Landa, Alberto Rodríguez-Lorenzana, Guido Mascialino, Juan Carlos Arango-Lasprilla, Erick Orozco-Acosta, Diego Rivera.

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
