## [Decision Letter · Decision Letter 0]

20 Aug 2025

Dear Dr. Orozco-Acosta,

We look forward to receiving your revised manuscript.

Kind regards,

Alejandro Botero Carvajal, Ph.D

Academic Editor

PLOS ONE

Journal Requirements:

Additional Editor Comments:

Please see the comments below.

Reviewer's Responses to Questions

**Comments to the Author**

1. Is the manuscript technically sound, and do the data support the conclusions?

Reviewer #1: Yes

Reviewer #2: Yes

2. Has the statistical analysis been performed appropriately and rigorously?

Reviewer #1: Yes

Reviewer #2: I Don't Know

3. Have the authors made all data underlying the findings in their manuscript fully available?

Reviewer #1: No

Reviewer #2: No

4. Is the manuscript presented in an intelligible fashion and written in standard English?

Reviewer #1: Yes

Reviewer #2: Yes

Reviewer #1: Summary Paragraph

The paper titled "Normative Data for Learning and Memory Test (TAMV-I) Based on Item Response Theory and Linear Mixed Models" evaluates the psychometric properties of the TAMV-I test using Item Response Theory (IRT) and Linear Mixed Models, generating normative data from a sample of 1,640 participants across three Latin American countries and Spain. The study aims to provide robust normative data and explore the interaction of sociodemographic variables with performance. Results suggest that variables such as age and country of origin significantly influence test performance, indicating a need to carefully interpret results based on these factors.

Overall, the study addresses an important issue in neuropsychological testing: the lack of access to standardized and high-quality neuropsychological assessments and the limitations of traditional statistical methods (e.g., Classical Test Theory, CTT) in generating reliable normative data. However, the discussion regarding country-specific effects and the variability in normative data interpretation remains underdeveloped. The study raises a critical question: If performance varies significantly across demographic factors, how should normative data be applied effectively in diverse populations?

Major Issues

1. Results are incomplete and difficult to interpret from figures.

• The presentation of results is unclear, making it challenging to extract key findings.

• Figures should be more structured and detailed, with improved resolution for readability. Add notes explaining the main features of figures.

2. Methodology

• Sampling Procedure:

o It is not explicitly stated how participants were selected.

o Was the sample size for each country predetermined, or was it subject to availability?

o What was the attrition rate (statistical death)?

• Data Collection:

o Were questionnaires filled out on paper or online?

3. Results

• Country-Specific Data Not Reported:

o The study claims that performance varies by country, but country-specific analyses are not shown in the results section.

o Adding a comparison table by country (e.g., means, SDs, effect sizes) would improve clarity.

• Normative Data Calculation:

o The example provided is useful, but the online calculator should be demonstrated visually.

o What does the calculator output look like? (e.g., a figure, table, or graph comparing the individual score to normative data).

o How was the calculator validated? Are there key parameters that need to be reported for its use?

4. Discussion

• Impact of Sociodemographic Variables:

o The authors mention that age, country, and gender influence test performance, but the implications of these interactions are not fully discussed.

o How should clinicians and researchers adjust their interpretations if these factors significantly affect results?

• Educational System Differences:

o The sample includes countries with diverse education systems (low-, middle-, and high-income settings).

o Could differences in educational background explain some of the performance variation?

o Would it be useful to control for education level as a covariate when establishing normative scores?

• Comparison to CTT, IRT, and Rasch Modeling:

o The study criticizes CTT but does not extensively discuss why IRT provides a significant advantage over traditional methods.

o Adding a brief theoretical discussion comparing CTT, IRT, and Rasch models would strengthen the paper.

Minor Issues

1. Title

• The title should specify the target population (e.g., "Normative Data for Learning and Memory Test (TAMV-I) in Latin America and Spain").

2. Introduction

• Line 90-91: There is a typo with an unnecessary period.

• Line 93: The author's last name should appear before the citation.

• Clarify Study Contribution:

o The introduction discusses the limitations of previous neuropsychological test standardization but does not explicitly state whether:

Prior normative data are unreliable.

The new approach offers an entirely different framework for interpretation.

The study only improves the statistical methodology without changing the core interpretation of the test.

o The authors should explicitly clarify how their results improve practical test interpretation.

3. Figures and Tables

• Figure 2:

o The resolution is too low, making it difficult to read.

o Consider replotting the figure with better formatting (e.g., larger font size, clearer legends).

Overall Assessment

The paper addresses an important issue in neuropsychological assessment—the need for better normative data using advanced statistical methods. The use of Item Response Theory (IRT) and Linear Mixed Models is a strong methodological choice that improves upon traditional Classical Test Theory (CTT).

However, several key aspects require improvement:

1. Results need clearer presentation—figures should be improved, and country-specific data should be explicitly reported.

2. Discussion should explore the implications of sociodemographic effects in more depth—especially regarding country, age, and education.

3. The normative data calculator should be better described—including its validation process and output format.

Recommendation: Major Revision Required

• The paper has substantial theoretical and methodological contributions, but unclear presentation and insufficient discussion weaken its impact.

• Addressing these issues will significantly improve clarity and practical applicability for clinicians and researchers using the TAMV-I test.

Reviewer #2: First, I would like to thank the esteemed researchers for their considerable effort in designing and conducting this study. I would like to offer a few suggestions to help improve the overall quality of the research:

1- The abstract lacks specificity, and there is no well-defined hypothesis or research question, making it difficult to understand the significance of the study. Without a clearly stated aim, it is challenging to assess the appropriateness of the methodology or the relevance of the findings. It should clarify the focus of the research, its importance, and the gap it intends to address or solve a problem or improve the way things are currently being done.

2- The writer has used certain non-mesh keywords.

3- In the final paragraph of the Results section (line 307), the symbols "@@@" are used, which are unclear. The same issue is repeated in the Discussion section at line 401.

4- Lines 408 to 412 state that normative data cannot be generalized to an entire region due to varying influencing conditions. Doesn’t this statement itself call into question the validity of the entire study, whose primary aim was to establish normative data?

5- In the Data Collection and Sampling Method section, it is unclear what measures were taken to ensure randomization. Robust sampling practices are essential for the generalization and reliability of findings, and the absence of details regarding randomization raises concerns about potential bias of the sample.

6- Some tables are missing units of measurement, have too many decimal places, or show numbers in inconsistent formats. These small details can make the data harder to follow and take away from the overall clarity and professionalism of the presentation. The quality of visual data presentation is substandard. Figures are at times too low in resolution, poorly labeled, or lack clear scale bars, keys, and legend information necessary for independent comprehension

7- There are no disclosures of potential conflicts of interest, funding sources, or data management protocols that would assure readers of the adherence to established ethical norms

Then major revision required prior to resubmission.

**Do you want your identity to be public for this peer review?** For information about this choice, including consent withdrawal, please see our Privacy Policy

Reviewer #1: **Yes:**  Cesar Acevedo-Triana

Reviewer #2: No

---

## [Author Response · Author response to Decision Letter 1]

4 Dec 2025

Reviewer #1

Overall, the study addresses an important issue in neuropsychological testing: the lack of access to standardized and high-quality neuropsychological assessments and the limitations of traditional statistical methods (e.g., Classical Test Theory, CTT) in generating reliable normative data. However, the discussion regarding country-specific effects and the variability in normative data interpretation remains underdeveloped. The study raises a critical question: If performance varies significantly across demographic factors, how should normative data be applied effectively in diverse populations?

Response: Thank you for your incisive comment. We think you have touched on a central issue we attempt to address in this study. Performance on neuropsychological measures does vary significantly across demographic factors, which presents an important challenge when attempting to develop representative norms. Traditionally, norms have been stratified by age groups and, less often, by other variables such as education or ethnicity/race. However, this approach suffers from two limitations: 1) we do not know a priori which variables are related to neuropsychological performance; and, 2) stratifying norms reduces power by limiting the amount of subjects in each category. In order to address these and other issues, we chose to use regression-based norming in this study. By doing so we can identify which variables are affecting performance beforehand and include them in the regression equation, which can transform direct scores to standardized values that account for demographic variables. Given that this approach identifies variables that affect performance, this variability is controlled for in the regression model, thus providing accurate norms for several demographic groups, including age, gender, education, and country related groupings. We added a brief explanation of this issue in the discussion, lines 470-473.

The presentation of results is unclear, making it challenging to extract key findings.

Response: We appreciate this observation. To improve the clarity and accessibility of the results, we have restructured the Methods and Results sections.

Figures should be more structured and detailed, with improved resolution for readability. Add notes explaining the main features of figures.

Response: Thank you for this suggestion. We have reformatted all figures to enhance readability by increasing resolution, enlarging font sizes, and improving legends. In addition, we have added explanatory notes on the figures to highlight their main features and guide interpretation (see line 479).

Sampling Procedure: It is not explicitly stated how participants were selected.

Response: We thank the reviewer for this observation. In the revised Procedure section (page 8), we have clarified the sampling process. Local research teams in each country first established agreements with schools and high schools to present the study. After obtaining authorization from the institutions, students and their families were informed about the project and invited to participate voluntarily.

Sampling Procedure: Was the sample size for each country predetermined, or was it subject to availability?

Response: The sample size for each country was subject to availability at the collaborating institutions rather than predetermined. However, local research teams ensured balanced distributions across sex and age groups, and mean parental education (MPE) was monitored in each subsample (see Table 1). This approach enhanced representativeness despite the convenience sampling strategy. Moreover, the final sample size allowed the calculation of sampling error, supporting the reliability of the normative estimates. Following Innocenti et al. (2023), and assuming a 95% confidence level, and with z₀ = –0.954, the expected standard error was 0.2679 for Spain, 0.3153 for Honduras, 0.2503 for Colombia, 0.2178 for Ecuador, and 0.1280 for the total sample. These values fall within acceptable ranges, confirming that the achieved precision is sufficient for the study aims and for the generation of robust and clinically meaningful normative data. This information was included in participants paragraph.

Sampling Procedure: What was the attrition rate (statistical death)?

Response: We thank the reviewer for this question. From the original sample of 1,748 participants, 108 cases were excluded due to incomplete data, resulting in a final sample of 1,640 participants. This corresponds to an attrition rate of approximately 6.2%, which is within acceptable limits for large-scale normative studies.

Data Collection: Were questionnaires filled out on paper or online?

Response: We appreciate this question. All questionnaires were completed on paper, in accordance with the administration guideline of the test (see page 7).

Country-Specific Data Not Reported: The study claims that performance varies by country, but country-specific analyses are not shown in the results section.

Response: We thank the reviewer for this observation. In the revised manuscript, we have clarified the country-specific effects in the Results section (page 13). In addition, we improved the visualization of these effects by replotting Figure 3, and table 2.

Adding a comparison table by country (e.g., means, SDs, effect sizes) would improve clarity.

Response: We appreciate this suggestion. However, we decided not to include a comparison table by country based on means and standard deviations. Such tables may lead readers to interpret raw averages as normative values, which can be misleading and have been repeatedly criticized in the neuropsychological literature. Our approach, combining Item Response Theory with Linear Mixed Models, provides more robust normative estimates adjusted for relevant covariates (e.g., age, sex, parental education, country), avoiding the limitations of unadjusted descriptive statistics. To maintain clarity, we emphasize the adjusted normative data and provide a practical calculator for clinical application.

Normative Data Calculation: The example provided is useful, but the online calculator should be demonstrated visually.

Response: We thank the reviewer for this helpful suggestion. In the revised manuscript, we included both the link to the online calculator https://diegorivera.shinyapps.io/calculator_tamvi_tri/, where readers can obtain their own computations, similar to the next screenshot.

Normative Data Calculation: What does the calculator output look like? (e.g., a figure, table, or graph comparing the individual score to normative data).

Response: We thank the reviewer for this observation. In the revised manuscript, we clarified that the calculator output is presented as a table displaying the individual’s ability score, z-score, and adjusted percentile. See last comment.

Normative Data Calculation: How was the calculator validated? Are there key parameters that need to be reported for its use?

Response: Thank you for this important question. The online calculator was validated by cross-checking its outputs against the normative conversions obtained directly from the statistical models (IRT [supplemental S1] and LMM [table 2]). For a set of randomly selected response patterns, the calculator’s ability scores, residuals, z-scores, and adjusted percentiles were compared with the values generated by R scripts, showing perfect correspondence. Regarding key parameters, the calculator requires the following inputs: item-by-item test responses, age, sex, parental years of education, and country. The outputs provided are (a) the individual’s ability score (θ), (b) the predicted ability score based on covariates, (c) the residual and standardized residual (z-score), and (d) the adjusted percentile. These parameters are explicitly reported in the revised Normative data application section to facilitate transparency and reproducibility.

Impact of Sociodemographic Variables: The authors mention that age, country, and gender influence test performance, but the implications of these interactions are not fully discussed. How should clinicians and researchers adjust their interpretations if these factors significantly affect results?

Response: Given that the use of regression-based norms already takes into account the influence of these factors, such as gender, age, and country, the clinician can feel sure that their interpretations are accurate once scores are transformed from raw data to standardized units by using the calculator.

Educational System Differences: The sample includes countries with diverse education systems (low-, middle-, and high-income settings). Could differences in educational background explain some of the performance variation?

Response: Thank you for your comment. We agree that education systems can vary by regional income differences. Given that we entered the country as a variable in the regression model, we believe these differences are largely controlled for.

Would it be useful to control for education level as a covariate when establishing normative scores?

Response: We agree with the reviewer that education is a crucial variable to consider when generating normative data. In our study, we included mean parental years of education (MPE) as a covariate in the Linear Mixed Models, since the participants themselves were children and adolescents. MPE is widely used in pediatric neuropsychology as a proxy for the child’s educational and cognitive environment, and it significantly influences test performance. Importantly, we did not use the child’s own years of schooling as a covariate, because this variable is highly collinear with age in this developmental range, which could bias parameter estimation. By modeling MPE alongside age, sex, and country, our approach provides normative scores that are sensitive to educational background while avoiding multicollinearity issues. This decision is described in the Methods section (Demographic effects) and further discussed in the manuscript.

Comparison to CTT, IRT, and Rasch Modeling:

The study criticizes CTT but does not extensively discuss why IRT provides a significant advantage over traditional methods.

Response: We appreciate this observation. In the revised Discussion section, we expanded the comparison between CTT, and IRT

Adding a brief theoretical discussion comparing CTT, IRT, and Rasch models would strengthen the paper.

Response: We thank the reviewer for this suggestion. In the revised version, we added a paragraph in the Introduction section providing a concise theoretical comparison of CTT, and IRT.

The title should specify the target population (e.g., "Normative Data for Learning and Memory Test (TAMV-I) in Latin America and Spain").

Response: We agree with this suggestion and have modified the title accordingly. The revised title is: “Normative Data for the TAMV-I in Latin American and Spanish Children: An Item Response Theory and Linear Mixed Models Approach.” This change highlights the target population and clarifies the scope of the normative data while retaining the methodological focus of the study.

Line 90-91: There is a typo with an unnecessary period.

Response: We thank the reviewer for noticing this detail. The sentence was rewritten to remove the unnecessary period and to improve the flow of the paragraph.

Line 93: The author's last name should appear before the citation.

Response: We thank the reviewer for pointing this out. The reference has been corrected so that the author’s last name appears before the citation

Clarify Study Contribution: The introduction discusses the limitations of previous neuropsychological test standardization but does not explicitly state whether:

• Prior normative data are unreliable.

• The new approach offers an entirely different framework for interpretation.

• The study only improves the statistical methodology without changing the core interpretation of the test.

Response: We appreciate this insightful comment. In the revised Introduction, we clarified that prior normative data are not regarded as unreliable; however, they present two major limitations. First, existing approaches do not consider the psychometric behavior of each item in terms of difficulty and discrimination across trials, but instead rely on simple univariate analyses. Second, norms are usually calculated separately for each score, implicitly assuming independence between trials, which is not clinically accurate. Our study addresses these gaps by examining the item-level psychometric properties within each trial and by modeling them jointly using Item Response Theory combined with Linear Mixed Models. This multivariate framework yields more precise and robust normative estimates and aligns more closely with the realities of clinical assessment, where inter-trial dependencies are central to interpretation.

The authors should explicitly clarify how their results improve practical test interpretation.

Response: We thank the reviewer for this valuable comment. In the revised Discussion, we explicitly highlight how our results improve clinical interpretation. By incorporating item-level parameters of difficulty and discrimination across trials, and by modeling these jointly through Item Response Theory and Linear Mixed Models, the normative data provide clinicians with more precise and covariate-adjusted percentiles. Unlike traditional univariate norms that treat each trial as independent, our approach reflects the inter-trial dependencies observed in clinical practice, thereby offering a framework that is both psychometrically robust and clinically meaningful. This allows practitioners to better distinguish between normal variability and true impairment, ultimately enhancing diagnostic accuracy and treatment planning.

Figure 2: The resolution is too low, making it difficult to read.

Figure 2: Consider replotting the figure with better formatting (e.g., larger font size, clearer legends).

Response: We appreciate this suggestion. Figure 2 (now Figure 3) has been replotted at higher resolution, with larger font size, clearer legends, and improved formatting to enhance readability.

Reviewer #2:

1- The abstract lacks specificity, and there is no well-defined hypothesis or research question, making it difficult to understand the significance of the study. Without a clearly stated aim, it is challenging to assess the appropriateness of the methodology or the relevance of the findings. It should clarify the focus of the research, its importance, and the gap it intends to address or solve a problem or improve the way things are currently being done.

Response: We thank the reviewer for this observation. The abstract has been revised to increase specificity and clarity.

2- The writer has used certain non-mesh keywords.

Response: We thank the reviewer for this observation. The keywords have been revised and updated to standardized MeSH terms. The final set of keywords included in the manuscript is: Neuropsychological Tests, Memory, Child, Adolescent, Cross-Cultural Comparison, Psychometrics, Statistical Models.

3- In the final paragraph of the Results section (line 307), the symbols "@@@" are used, which are unclear. The same issue is repeated in the Discussion section at line 401.

Response: We thank the reviewer for noticing this error. The symbols “@@@” were unintentional and have now been replaced in the manuscript with the reference to the online calculator: https://diegorivera.shinyapps.io/calculator_tamvi_tri/

4- Lines 408 to 412 state that normative data cannot be generalized to an entire region due to varying influencing conditions. Doesn’t this statement itself call into question the validity of the entire study, whose primary aim was to establish normative data?

Response: We thank the reviewer for this observation and understand the concern. Our intention was not to suggest that the normative data generated in this study are invalid, but rather to emphasize an important clinical caution. Normative data provide robust and clinically useful references for the countries included; however, as with any large-scale normative study, they should not be assumed to account for all poss

---

## [Decision Letter · Decision Letter 1]

5 Jan 2026

Normative Data for Learning and Memory Test (TAMV-I) in Latin American and Spanish Children: An Item Response Theory and Linear Mixed Models Approach

PONE-D-25-02804R1

Dear Dr. Orozco-Acosta,

We’re pleased to inform you that your manuscript has been judged scientifically suitable for publication and will be formally accepted for publication once it meets all outstanding technical requirements.

Kind regards,

Alejandro Botero Carvajal, Ph.D

Academic Editor

PLOS One

Additional Editor Comments (optional):

Reviewers' comments:

Reviewer's Responses to Questions

**Comments to the Author**

Reviewer #1: All comments have been addressed

2. Is the manuscript technically sound, and do the data support the conclusions?

Reviewer #1: Yes

3. Has the statistical analysis been performed appropriately and rigorously?

Reviewer #1: Yes

4. Have the authors made all data underlying the findings in their manuscript fully available?

Reviewer #1: No

5. Is the manuscript presented in an intelligible fashion and written in standard English?

Reviewer #1: Yes

Reviewer #1: I have reviewed the revised manuscript and find that the authors have responded appropriately to the reviewers’ comments, with clear improvements in clarity and presentation. I recommend accepting the manuscript as revised.

**Do you want your identity to be public for this peer review?** For information about this choice, including consent withdrawal, please see our Privacy Policy

Reviewer #1: **Yes:**  Cesar Acevedo-Triana

---

## [Editor Report · Acceptance letter]

PONE-D-25-02804R1

PLOS One

Dear Dr. Orozco-Acosta,

I'm pleased to inform you that your manuscript has been deemed suitable for publication in PLOS One. Congratulations! Your manuscript is now being handed over to our production team.

Kind regards,

on behalf of

Dr. Alejandro Botero Carvajal

Academic Editor

PLOS One